# The Innate Rates of Increase in Two Common Stored Grain Insects under Different Grain Storage Conditions and Times

**DOI:** 10.3390/insects15060441

**Published:** 2024-06-11

**Authors:** Fuji Jian

**Affiliations:** Department of Biosystems Engineering, University of Manitoba, Winnipeg, MB R3T 5V6, Canada; fuji.jian@umanitoba.ca

**Keywords:** stored grain, carrying capacity, prediction, population dynamics, influencing factors, grain volume

## Abstract

**Simple Summary:**

Comparison between innate rate of increase values (*r*) and identifying carrying capacity are critical because this comparison and identification can identify the effect of ecological factors on the development and multiplication of insects and mites in stored grain bins. After this comparison and identification, the insect population can be predicted using the identified *r* values. In the literature, the *r* value has been measured by determining the instantaneous birth rate and instantaneous death rate. This measured *r* value cannot be used because it does not represent grain storage conditions. This study used a continuous time analysis method to calculate the *r* values for *Tribolium castaneum* and *Cryptolestes ferrugineus* by using the data published in the literature. In the literature, the tested grain temperatures covered the suboptimal and optimal temperatures for the two insect species. Both constant and fluctuating temperatures were used. The grain bulk size varied from 0.3 to 14 kg of food in long vertical columns or shallow containers. The grain was whole wheat or whole wheat plus cracked wheat with a moisture content of 13.8 to 14.5%. These calculated *r* values could be used to predict the insect population dynamics in stored grain bins.

**Abstract:**

The determination of innate rate of increase (*r*) values under different grain storage conditions is critical for insect population predictions. The *r* values for *Cryptolestes ferrugineus* (Stephens) and *Tribolium castaneum* (Herbst) were calculated by using a new suggested method (continuous time analysis) and data from the literature, and these calculated *r* values were compared to identify the *r* values and carrying capacities under real grain storage conditions and times. The insects were reared in small glass vials (0.3 kg wheat), small PVC columns (2 kg wheat), large PVC columns (14 kg wheat), and shallow containers (14 kg wheat or wheat + cracked wheat). The wheat or cracked wheat had 13.8 to 14.5% moisture contents at different constant temperatures (17.5 to 42.5 °C) and fluctuating temperatures. The *r* values at the beginning of the population were the highest. Before *r* became negative, it gradually decreased with increasing time. After the *r* value became negative, it sometimes increased to positive; however, the rebounded *r* was much less than the initial *r* and gradually tended to stabilize within an up-and-down range. This up-and-down *r* was related to the carrying capacity. The larger the grain bulk, the higher the innate rate was for both species. The *r* values associated with 14 kg of wheat could be used to predict the insect population dynamics in stored grain bins.

## 1. Introduction

Ectothermic insects can have high assimilation efficiencies because they need to use the minimum energy to regulate their body temperature (their body temperatures are the same as their environments). So, most of the energy gained by individual insects is directly used to produce eggs and to grow their body mass. Stored grain insects and mites consume grain to meet their energy requirements, and each individual insect or mite only needs one or a few grain kernels for their entire life due to their high assimilation efficiency and their small body sizes. The entire life cycle of stored grain insects is usually less than one month, and stored grain mites take only 9 to 11 d to complete their life cycle under the optimal temperatures and moisture conditions [1]. For the above reasons, insects and mites have a high fecundity rate (laying a huge number of eggs) and an exponential population increase in a short time under their optimum multiplication conditions. Under the optimum conditions, most stored product insects can multiply more than 20 times in one month. For example, three pairs of *Tribulitum castaneum* can become 10,000 insects in 20 weeks at 30 °C [2]. Mites are smaller than insects; hence, mites usually have a higher reproduction rate than insects, and their reproduction rates can be more than 2500 times/month [3]. Their high reproduction rates are a big threat and menace to stored grain and beget challenging work in stored grain management. Their reproduction rates are influenced by many factors, such as temperature, grain moisture content, insect density, and food source. Therefore, the prediction of their population under different storage conditions is required to develop the optimal strategy for insect and mite control.

Insect populations can be predicted by calculating their instantaneous rate of increase per unit time (usually a generation or a life cycle) without considering the effect of density (Equation (1)).
(1)$$r_{m}=b-d$$
where *r_m_* is the intrinsic (innate) rate of increase per unit time per capita (referred to as the geometric growth rate of increase, decimal), *b* is the birth rate per unit time per capita (decimal), and *d* is the death rate per unit time per capita (decimal). The assumption of this method is not strictly true but provides a theoretical foundation and approximation [4]. The model developed using this method is termed a geometric (discrete time) model of population growth. If *b* and *d* change continually, the population will have an exponential increase, and the population can be predicted using the exponential model (Equation (2)) [5].
(2)$$N_{t}=N_{0}e^{rt}$$
where *N_t_* and *N*_0_ are the insect density (insects/kg of wheat) at time *t* and time zero, respectively; *t* is the time elapsed from time zero to *N_t_*; and *r* is the instantaneous rate of increase or the intrinsic rate of increase (referred to as the exponential growth rate of increase, decimal).

The relationship between *r_m_* and *r* is:(3)$$1+r_{m}=e^{r}$$

For a population with a stable age distribution, researchers have assumed *r_m_* = *r* [4,6,7]. Both *r_m_* and *r* can be a positive or negative value, which indicates an exponential increase or decrease in the population, respectively, while 0 indicates no change in the population. This assumption has certain errors because exponential growth models have a faster growth rate than geometric models, so the population size of exponentially growing populations outpaces geometrically growing populations over time. This error is small in a short period (a few generations).

Based on Equation (2) and if the insect density is evaluated at different times, the innate rate of increase at different counting times can be determined as:(4)$$r=\frac{\ln(\frac{N_{t+1}}{N_{t}})}{t}$$
where *N_t_*_+1_ and *N_t_* are the insect density at times *t* + 1 and *t* (insects/kg of wheat), respectively.

When Equation (1) is used, *r_m_* can be measured by determining the instantaneous birth rate and the instantaneous death rate. This is usually measured by rearing one or a few pairs of insects or mites. All the produced offspring are removed during the rearing period, and the birth rate per unit time per capita (*b*) and the death rate per unit time per capita (*d*) are calculated by averaging over the individuals in the cohort. The *r_m_* value is determined after *b* and *d* are estimated. This determination is used extensively by applied ecologists such as stored grain entomologists as a means of describing the growth potential of an organism. The *r_m_* values of most stored grain insect and mite species have been determined under lab conditions in a small amount of grain (usually less than 0.1 g) using this method [4,8,9,10,11]. These determined *r_m_* values are the values under the ideal determined conditions of temperature and relative humidity because the multiplication and development of the insects under the rearing conditions are not disturbed by other insects and not limited by food sources, variable environmental conditions (such as variations in temperature and moisture content), or the chemical environment.

The values of the exponential growth rate of increase (*r*) can be determined by using continuous time analysis methods such as rearing a few pairs of insects in a large amount of grain (e.g., more than a few kg of grain) without removing their offspring. After one or several generations (life cycles) under the tested conditions, the insect population can be determined by counting all the offspring and adults at different times. Therefore, different *N_t_*_+1_ values can be determined for different generations (at different times). Under this condition, the calculated *r* value at the beginning of storage may be different from *r* later during storage because (1) *r* at the beginning of storage is mostly associated with the parent insects, and the competitive effect at the very beginning of the population may be low due to the low insect density; (2) after a few generations, insect stages can overlap with each other, and a competitive effect can play a critical role with an increase in the insect number; (3) the food source and quality can gradually decrease along with insect growth and multiplication, and (4) after insect infestation, the chemical environment can be modified (become unsuitable for insect multiplication). Therefore, the condition used to determine the *r* value is closer to the real storage conditions than the determined *r_m_* value because the conditions for the measurement of *r* can include the effect of competition among individuals, limitations in food and space, patch size (grain volume), and the effect of a modified chemical environment. This calculated *r* presents the different innate rates of increase at different generations, along with an increase in the storage time. However, to the best of the knowledge of the author, there is no published innate rate of increase value (*r*) calculated using this method for grain storage insects and mites, and using the *r_m_* value might result in a wrong prediction of the insect population for the above reasons. The difference between the measured *r_m_* and *r* values is not known. It is not known which value should be used to predict the insect population dynamics in stored grain bins.

Even though insects and mites can have high fecundity rates, their populations are dynamic because the population number can be influenced by many factors, such as weather (grain temperature and moisture content), food sources (different grain types and food quality), patch size (space, grain volume), natural enemies, and insect disease. Food and space availability can affect the size that an insect population can attain. This concept is referred to as the carrying capacity (*K*) of an environment. Carrying (density) capacity has been defined as a population’s equilibrium density, which means that the realized rate of per capita increase at the time is 0. The following logistic equation is the most common formulation for the description of population growth and the relationship between *r* and *K* [12]:(5)$$\frac{dN}{Ndt}=r-r\frac{N}{K}\;or\;\frac{dN}{dt}=rN(1-\frac{N}{K})$$
where *N* is the insect density at time *t*, *K* is the carrying capacity (the same unit as *N*), and *r* is the innate rate of increase [12]. Equation (5) indicates that *r* = $\frac{dN}{Ndt}$ when *N* is very small (much smaller than *K*), and high values of *N* tend to have lower values of *r*. Therefore, *K* is not independent from *r*. There is no report on the *K* value for stored grain insects, and it is not known whether stored grain insects have a carrying capacity. The relationship between the carrying capacity and the innate rate of increase has not been established.

To predict insect populations with a high accuracy, there is a trend in population models toward the incorporation of greater biological detail into the descriptions of the life history processes responsible for population changes. These biological detail models usually incorporate the age and stage structure of the population. Geometric and exponential models are biologically oversimplified because they assume that all individuals in the population are the same age and breed continuously. Variable rates of reproduction, survival, and resource utilization are ignored in both models. However, the condition used to determine *r* is close to the real grain storage condition. These determined *r* values can be directly used to quickly estimate the insect population.

Comparison between *r_m_* and *r* and identifying *K* are critical because this comparison and identification can identify the effect of ecological factors on the development and multiplication of insects and mites in stored grain bins. Addressing these research gaps, the objective of this study was to calculate *r* using the available lab data from the literature and compare these values with the reported *r_m_* values. After this calculation and comparison, the carrying capacity was found. The main factors influencing the innate rate of two major stored grain insect species were identified, the trend in the *r* values was discovered, and *r* values which could be used to predict the insect population in real storage bins were determined.

## 2. Materials and Methods

### 2.1. Data Used

Three original sources of data which could be used to calculate the innate rate of the rusty grain beetle, *Cryptolestes ferrugineus* (Stephens), and the red flour beetle, *Tribolium castaneum* (Herbst), using the suggested continuous time analysis method were identified. The rusty grain beetle and the red flour beetle are in the families of Laemophloeidae and Tenebrionidae, respectively, and both are in the order of Coleoptera. The three data sources are Jian et al. [13], Tripathi et al. [2], and Jagadiswaran et al. [14]. The data reported by Jian et al. [13] and Tripathi et al. [2] were collected at 21, 25, 30, 35 °C, Decrease T, and Increase T in wheat with a 13.8 to 14.5% moisture content (on a wet basis). Decrease T was set at 30 °C at the beginning and in the first 4 wks and then decreased by 1 °C/wk until the temperature was 10 °C, decreased by 5 °C/wk until the temperature was −15 °C, and kept at −15 °C for 3 wks. Increase T was set at 21 °C at the beginning and in the first 2 wks and then increased by 1 °C/wk until the temperature was 38 °C and kept at 38 °C for 5 wks. The wheat bulks (patches) used were categorized as small (glass vial with 50 mL inner volume filled with less than 0.5 kg of wheat, referred to as Glass vial), medium (small PVC column with 0.15 m diameter and 15 cm high, total 2.6 L inner volume filled with 2 kg of wheat, referred to as Small PVC), and large (large PVC column with 0.15 m diameter and 102 cm high, total 18 L inner volume filled with 14 kg of wheat, referred to as Large PVC) (Figure 1). The number of adults and offspring in the infested wheat in the different grain bulks were counted every 4 wks for up to 31 wks. The innate rates at the counting times (referred to as storage time) were calculated. The data reported by Jagadiswaran et al. [14] were collected at 30 °C in large PVC columns and shallow containers (460 mm long, 660 mm wide, and 150 mm high; referred to as SC). The containers were filled with 14 kg of wheat or diet (whole wheat and cracked wheat at a 19:1 ratio by mass) with a 14.5% moisture content. The experimental procedure conducted by Jagadiswaran et al. [14] is similar to that conducted by Jian et al. [13] and Tripathi et al. [2]. The experiments conducted by Jagadiswaran et al. [14] on the diet and whole wheat were referred to as Diet and Whole, respectively. For example, SC-Diet and SC-Whole were the experiments conducted in the shallow containers filled with 14 kg of diet and 14 kg of whole wheat, respectively. The equilibrium relative humidity (RH) of the 13.8 to 14.5% moisture content wheat was 60 to 70%. All the data used in this study had a similar relative humidity (RH). Single insect species were introduced into each patch, and the insect number changed during the test period (maximum 217 d).

Two original data sources which could be used to determine the *r_m_* values for the rusty grain beetle and the red flour beetle were identified. The two data sources are Smith [4] and White [7]. The data related to the rusty grain beetle were collected from 17.5 to 42.5 °C at 2.5 °C intervals, at 70% RH, and in 0.5 g of wheat flour plus 5% wheat germ (by mass). The time to hatching and the percentage hatch of eggs were observed under the tested conditions. Each larva was reared individually in 0.5 g whole wheat flour plus 5% wheat germ (by mass) in a glass vial (5 cm diameter and 1.25 cm high). These vials were examined daily until the adults emerged. To observe the oviposition, each pair of male and female adults was placed in a glass vial on 0.5 g white flour plus 5% wheat germ (by mass). The oviposition rate of these beetles was determined at the tested temperature and humidity at which they had completed larval and pupal development [4]. The data related to the red flour beetle were collected from 20 to 37.5 °C, at 25 to 65% RH, and in less than 1 kg of whole wheat [7]. The time to hatching and percentage hatch of eggs were observed for batches of 100 eggs under the tested conditions. The mortality of the larvae and pupae were determined in glass vials containing 50 wheat kernels. The dates of pupation and adult emergence were recorded daily. The number of eggs laid was assessed indirectly by estimating fertility (number of live progeny) and dividing this value by the measured survival rate of immatures. During the rearing process of the rusty grain beetle and the red flour beetle, the mortality for each stage was determined. These measured survival rate and oviposition rates were used to calculate the *r_m_* values. The authors of [4,7] reported the estimated *r_m_* values.

### 2.2. Calculation of r and Carrying Density

The *r_m_* values reported in the literature were directly used, and the unit of the *r_m_* values was converted into days^−1^. *r* was calculated using Equation (4), and the calculated *r* values were 0 to 28, 28 to 56, 56 to 84, 84 to 112, 112 to 140, and 140 to 168 d, which were reported at 0, 28, 56, 84, 112, and 140 d, respectively. The authors of [2,13,14] reported the live and dead adult and offspring numbers for each tested condition every four weeks. Only the reported live insect numbers were used in this study to calculate the insect densities and *r*. The initially introduced insects in all these studies were three pairs of adults less than 10 d old. For example, the *r* value for the first four weeks for *T. castaneum* at 21 °C in a large PVC column = $\frac{ln(\frac{total\;live\;insects\;at\;4\;wk\;and\;21\;℃\;in\;the\;Large\;PVC}{6})}{4\times 7}$, where 6 is the number of initially introduced insects, and 4 × 7 are the total days during this period.

Based on the definition of carrying capacity, the *K* value is the highest density at *r* = 0. This study did not find 0 in the calculated *r* values, and the *r* values in the first 4 wks were the highest (but density was not the highest) and gradually decreased to negative under all the tested conditions. Therefore, the *K* value was found by evaluating the innate rate value and insect density. If the current innate value was larger than both the previous one and the later one (the previous one was negative and the later one could be negative or positive, so *r* was close to 0), the highest density (density of adults plus offspring and density of adults) associated with the current innate value was the *K* value found. If more than two *K* values were found for one tested condition, the highest density was reported in this article.

The insect numbers associated with the experiments Decrease T and Increase T were also used to calculate the *r* values under the conditions of fluctuating temperatures. The calculation method was the same as that used for the constant temperatures.

To find whether the calculated innate rate in large PVC columns was similar to that inside grain bins, *r* was calculated by using the data reported by Abdelghany and Fields [15]. Abdelghany and Fields [15] introduced 600 adult rusty grain beetles into three barrels holding 300 kg of wheat each. The barrels were kept at 30 ± 3 °C for 5 mos. After 5 mos, the wheat was cooled to below 0 °C for 7 d, so the population growth was stopped, and the insects were counted.

### 2.3. Data Analysis

One-way ANOVA (Tukey’s test or Student’s *t*-test) was conducted to compare the *r* values under different test conditions and between the two species. For example, the calculated *r* values in the large PVC columns and the barrels associated with the data reported by Abdelghany and Fields [15] were compared by using Student’s *t*-test. Tukey’s test was conducted to find the difference among the carrying capacity at different temperatures or in different wheat bulks. All the datasets passed the normality tests in this study (Shapiro–Wilk test, Kolmogorov–Smirnov test, and mean with SD).

To find the relationship between the innate rate and previous insect density, correlation was conducted between the calculated innate rate and the previous insect density. For example, the calculated innate rate at 8 wks was correlated with the density of adults or adults plus offspring at 4 wks using Pearson’s Product–Moment correlation method. The correlation coefficient was used to evaluate the relationship, so the following question could be answered: whether the previous density influences the innate rate under grain storage conditions at different stages of population dynamics. SigmaPlot v13 (Systat Software, Palo Alto, CA, USA) was used to conduct the statistics.

## 3. Results

### 3.1. Innate Rate at the Beginning of the Population Dynamics (About 1 Month)

In 4 wks, the *r* values of *C*. *ferrugineus* in the large PVC columns were 0.082 ± 0.016, 0.121 ± 0.007, 0.151 ± 0.001, and 0.215 ± 0.008 days^−1^ at 21, 25, 30, and 35 °C, respectively. The reported *r_m_* values of C. *ferrugineus* were smaller than *r* in the large and small PVC columns, while they were higher than that in the glass vials (Figure 2). *r* in the large PVC columns was similar or slightly higher than that in the small PVC columns, while r in the glass vials was always lower than that in the larger grain containers (Figure 2). This result indicated that *r* of *C*. *ferrugineus* in stored grain bins could be similar to the calculated *r* value in the large or small PVC columns, and the reported *r_m_* could not be used to estimate the population of *C*. *ferrugineus* in real stored grain bins.

In 4 wks, *r* for *T. castaneum* in the large PVC columns was 0.086 ± 0.012, 0.130 ± 0.002, 0.138 ± 0.004, and 0.126 ±0.002 days^−1^ at 21, 25, 30, and 35 °C, respectively. For *T. castaneum* at 30 and 35 °C, the reported *r_m_* values were similar to or higher than *r* in the glass vials and large and small PVC columns, while at 21 and 25 °C, they were lower than that in the large and small PVC columns (Figure 3). *r* in the large PVC columns was always similar to or higher than that in the small PVC columns and higher than that in the glass vials for both insect species (Figure 2 and Figure 3). *r* for *T. castaneum* in the large PVC columns (14 kg of wheat) was double that in the small PVC columns (2 kg of wheat) at 21, 25, and 35 °C (Figure 2). This result indicated that *r* for *T. castaneum* in stored grain could be higher than the calculated *r* value in the large PVC columns, and the reported *r_m_* could not be used to estimate the population dynamics of *T. castaneum*. For both species, the size of the grain bulk influenced the *r* values, hence the insect multiplication.

In 4 wks and at 30 °C, the *r* value associated with the data reported by Tripathi et al. [2] was not significantly different from that when the data reported by Jagadiswaran et al. [14] were used (Tukey’s test, F = 5.061, *p* = 0.052, N = 3) (Figure 4). Tripath et al. [2] and Jagadiswaran et al. [14] used the same strain of insect, and the only difference between these two studies was the different wheat, with the same temperature and moisture content. Different proportions of cracked wheat between these two studies exist, which might result in variation in the determined *r* values (Figure 4). The higher *r* associated with SC-Diet (whole and cracked wheat) compared to that in SC-Whole (whole wheat) supported this conclusion (Figure 4). The larger surface area of the shallow container might be another reason resulting in the higher means of *r* in SC-Whole than those in the large PVC columns (Figure 4). The maximum *r* value for *T. castaneum* was about 0.18 days^−1^, and this maximum value was achieved in the shallow container filled with 5% cracked wheat (Figure 3 and Figure 4). Therefore, *T. castaneum* could have the highest *r* at the surface of stored grain bins.

### 3.2. Innate Rate of Overlapped Generations (After 1 Month)

The *r* value at less than 1 mo. of storage was the highest innate rate value for all the tested conditions (Figure 2, Figure 3 and Figure 4). Before *r* became negative, the *r* value gradually decreased with increasing storage time at any constant temperature in 13.8 to 14.5% MC wheat and for both species (Figure 2, Figure 3 and Figure 4), regardless of the insect density. After the innate rate became negative (population decreased), the innate rate could increase; however, the rebounded innate rate would not increase more and gradually tended towards stability in an up-and-down range (Figure 2, Figure 3 and Figure 4). These up-and-down innate rates were related to the carrying capacity of the tested conditions. This trend was more obvious inside the large and small PVC columns than that inside the glass vials (Figure 2, Figure 3 and Figure 4), and there was no difference in the innate rate between the large PVC and small PVC columns (Figure 1, Figure 2 and Figure 3). For example, at 35 °C, the range of the innate rate for *C. ferrugineus* was from 0.02 to −0.02 days^−1^, and for *T. castaneum*, it was from 0.03 to −0.03 days^−1^. The range of the innate rate for *T. castaneum* at 30 °C was similar to that at 35 °C (Figure 3).

When the data reported by Abdelghany and Fields [15] were used, the innate rate for *C. ferrugineus* was 0.056 ± 0.025 days^−1^, and the insect (live adults and offspring) density was 900 ± 400 insects/kg of wheat. The innate rate for rusty grain beetles in the large PVC columns at 30 °C in 5 mos. was 0.043 ± 0.001 days^−1^. There was no significant difference between these two calculated innate rates (Student’s *t*-test, *t* = 0.52, *p* = 0.631, and N = 3). Therefore, the innate rate calculated using the data collected in the large PVC columns could be used in stored grain bins.

Pearson’s correlation coefficients between the insect density and the innate rate values were negative (Table 1). Therefore, a higher density could result in a lower and negative innate rate. This was expected; however, there were only 16 out of 72 cases showing a significant correlation between the innate rate and insect density, in which *C. ferrugineus* represented 4 out of 36 cases and *T. castaneum* 12 out of 36. This result indicated that the innate rate for *T. castaneum* was more strongly related to its density than that for *C. ferrugineus,* and other factors, such as insect behaviors in different-size wheat patches and different insect age structures (such as stage overlaps), also influence the innate rate. Pearson’s correlation coefficients associated with the density of both adults and offspring were higher than those associated with adult density, except for 4 out of 72 cases (Table 1). Therefore, the total insect density, not only that of the adults, influenced the insect population growth.

Under fluctuating (decreased T or increased T) temperatures, the reported *r_m_* had different trends to the calculated innate rates, while *r* at fluctuating temperatures changed along with the change in the temperature (Figure 2 and Figure 3). The *r* values for the insects in different wheat containers were significantly different (Table 2), and *r* under fluctuating temperatures was similar to *r* at a constant temperature (equivalent to the average of the fluctuating temperatures). Therefore, the reported *r_m_* could not be used in stored bins with fluctuating temperatures, *r* could be used for stored grain, and stored grain bins could have larger *r* values.

### 3.3. Carrying Capacity

After the exponential multiplication in the first one or two months, the insect density decreased to lower than the carrying capacity; then, the insect density could bounce back, but the carrying capacity (density) was stable after 2 or 3 months (Figure 2, Figure 3 and Figure 4). After that period, the population of both species had a decreasing trend. At 21 and 25 °C, *K* in a small wheat container was significantly higher than that in a larger wheat container for both species (Table 3). At 30 and 35 °C, there were no significant differences among different wheat containers (Table 3). In a wheat column of the same size, the *K* values significantly increased along with an increase in the temperatures for both species, especially from 30 to 35 °C. This result indicated *K* was different at different temperatures and in different wheat volumes. The higher the grain temperature, the higher the *K* value, but the *r* values at *K* were lower than the maximum *r* value. When the temperature was ≤25 °C, the insects could survive at a high insect density in a small grain container, but the grain container did not influence *K* at ≥30 °C. Considering the huge volume of the grain bins, therefore, *K* inside bins might be smaller than that determined in the large PVC columns.

### 3.4. Differences in r and Carrying Capacity between Species

Under the same conditions, the innate rate for the *T. castaneum* parents was similar to that for *C. ferrugineus,* except at 35 °C (Figure 2 and Figure 3). At 35 °C in the large PVC columns, *r* for *C. ferrugineus* after one month of storage was significantly higher than that for *T. castaneum* (Student’s *t*-test, *t* = 10.967, *p* < 0.001, N = 3). However, the innate rate of the overlapped generations of both species was similar (Figure 2 and Figure 3). The only difference was their carrying capacity. The carrying capacity of *T. castaneum* was smaller than that of *C. ferrugineus* at any temperature in the large PVC columns. Therefore, the multiplication of *T. castaneum* was more influenced by the insect density and grain bulk size than that of *C. ferrugineus.*

## 4. Discussion

The intrinsic (innate) rate of (natural) increase is a measure of the rate of increase in populations that reproduce within discrete time intervals and possess generations that do not overlap when the population is free of competition with other individuals. This is also known as the Malthusian parameter [16]. Stored product insects usually have generation overlaps and stage overlaps. Some species, like the red flour beetle, have cannibalistic behavior, while rusty grain beetles rarely eat each other. The innate rate is also influenced by the amount of the grain bulk, space, patch size, food quality, and the insect density inside the grain bulk. Therefore, the reported *r_m_* does not represent the real innate rate of insects in stored grain bins. This study verified this conclusion because *r* for the rusty grain beetle in the large PVC columns was always larger than the reported *r_m_*, while *r* for the red flour beetle in the large PVC columns at 30 and 35 °C and in the first month of storage was always lower than the reported *r_m_*. The calculated *r* after 2 mos. was always smaller than that at 1 mo. Therefore, there might be a larger error when the reported *r_m_* is used to predict the insect population in stored grain bulks.

The dynamics of populations are usually described by the intrinsic rate of increase and the carrying capacity of a population. These two parameters are mostly used to describe different modes of life because fast population growth rates might result in poor competitive abilities (*r*-strategists), while *K*-strategists have slow-growing populations but may be superior competitors or efficient in using resources [17]. The reason behind these two strategies is that it is difficult to have both fast growth and be efficient in the use of resources at the same time; therefore, *r* and *K* strategies are often expected to be traded off against each other. However, different *r–K* relationships are observed, and the *r–K* distinction is not regarded as universal [18]. This study found that both species had a similar innate rate, but the red flour beetle had a lower *K* value than that of the rusty grain beetle. The population trends in these two common stored grain insect pests cannot be only explained by competition and efficiency in using resources.

Equation (5) describes the relationships between *r*, *K,* and *N*. After calculating the values of $\frac{dN}{Ndt}$, linear regression can be conducted to find *r* and *K*. $\frac{dN}{dt}=rN(1-\frac{N}{K})$ can be calculated by using polynomial regression. Therefore, linear regression was conducted to find this relationship (not reported in this article). However, the regressed equation had a low R^2^ (less than 0.35) and large bias residues. The regressed *r* and *K* values for most of the tested conditions had infinite standard errors, which indicated the values were unsuitable. The regressed *r* values were much higher or lower than the calculated *r* values reported in this article. The *K* values in some cases were 0, which is unreasonable. The basic assumption of Equation (5) is that *r* is a constant after reaching *K*. The calculated *r* values were not constant because the population of both species crashed (decreasing trend in the population) after reaching *K*, and the *r* values at the beginning of the population were the maximum and did not reach *K*. Therefore, the regression method was not suitable and not used to find *r* and *K* in this study.

At the beginning of grain storage, there will be maximum growth, as the food quality and quantity are high, competition is low, and the chemical environment of the stored grain is not changed and is under the ideal conditions for insect multiplication. The two species do not reach their carrying capacities in less than 3 months. As the insect numbers rise, there will be a reduction in the innate rate due to a decline in food quality, a change in the chemical environment, and an increase in competition. This expectation was observed in this study even under the Increase T (21 °C in the first 2 wks, then increased by 1 °C/wk) condition (Figure 2 and Figure 3). Under the Increase T condition and along with an increased temperature, the *r* values gradually decreased but did not become negative until the temperature was higher than 39 °C. This was contrary to our expectation because insects should have higher multiplication and development when the temperature is increased from 21 to 35 °C [4,7]. This gradually decreased *r* indicated that the insect population reached its carrying capacity. Therefore, the grain conditions (no infested grain) and insect density at the beginning of the population dynamics are the main factors influencing insect population dynamics.

This study also found that the decreased innate rate gradually tended toward observing a range of values, up and down within the range, and both species had carrying capacities associated with each tested condition. These up-and-down values for the innate rate and carrying capacity were not influenced by a decline in food quality or a change in the chemical environment. This study only observed the population of the insects in 168 d, and the insects could not move out of the grain containers. If insects were allowed to move out the grain container (e.g., to new places inside a bin), the migrated insects would multiply in a new place and would have a higher innate rate value. The unmigrated insects could also have a higher innate rate because of the decrease in competition after insect migration. Therefore, insects could have a higher innate rate (might be close to the maximum r) in stored grain bins than that determined in small grain bulks under lab conditions due to the large volume of bulk stored grain.

Jian et al. [19] concluded that the first two main factors influencing the population dynamics of the rusty grain beetle were the temperature and the previous insect numbers. This study found the total insect density, not only the adult density, influenced the insect population growth. The carrying capacity was different at different temperatures and in different wheat bulks. The higher the grain temperature, the higher the carrying capacity and the larger the innate rate, but the *r* values at the carrying capacity were lower than the maximum *r* value and were close to 0, in most cases being negative. Both species had different carrying capacities at different temperatures and in different grain containers. These different carrying capacities were more obvious between 35 °C and any other temperature. Under their optimum relative humidity and at 25, 30, and 35 °C, the development times for *C. ferrugineus* are about 40.6, 26.6, and 19.6 days [4], respectively, while *Tribolium castaneum* needs 29.9 ± 0.31, 18.4 ± 0.21, and 15.5 ± 0.33 days [7], respectively. These results indicate that the red flour beetle has a lower carrying capacity and a shorter life cycle than the rusty grain beetle. After reaching its carrying capacity, the population crashed (the density generally decreased, even though there might have been an increase in some cases). Therefore, the red flour beetle reaches its carrying capacity more quickly than the rusty grain beetle and will migrate to other locations in stored grain bins to avoid cannibalism [7]. These conclusions are explained by the fact that after harvest and before grain temperature is lower than 15 °C (due to the winter), (1) the high growth rate is the main factor resulting in high insect damage of grain kernels in this short period in western Canada; (2) insects emigrate to avoid reaching the carrying capacity and intraspecies competition; (3) the red flour beetle prefers migration to near the surface because the surface is convenient for the migration of insects with a larger body size [20], while the rusty grain beetle prefers moving inside the grain bulk due to its small body size [20]. These facts also explained that both are common pests in stored grain in the world, and both prefer different locations for multiplication.

## 5. Conclusions

In this study, the values of the innate rate of increase (*r*) for two common stored grain insect species were calculated, and these calculated *r* values were strongly related to the storage time and the grain storage conditions, such as grain temperature, moisture content (RH), insect density, grain bulk size (volume), and food source. The evaluated grain temperatures covered the suboptimal and optimal temperatures for the two insect species. The temperatures were constant or fluctuated. The grain bulk size varied from 0.3 to 14 kg of food in long vertical columns or shallow containers. The grain was whole wheat or whole wheat plus cracked wheat. It was found that the reported innate rates in the literature (*r_m_*), which were determined in small glass vials with less than 0.3 kg of food, could not be used to predict the insect population in stored grain bins. The exponential growth rate of increase (*r* values) determined in glass vials was lower than that in larger grain volumes and containers. The larger the grain bulks, the higher the *r* value was for both *C. ferrugineus* and *T. castaneum*. However, 2 kg of wheat was enough for *C. ferrugineus* to achieve its highest *r*, and *T. castaneum* needed more than 14 kg of wheat to achieve its highest value. Therefore, *r* values measured in a small amount of grain cannot be used to predict the insect population in stored grain bins. The innate rate calculated using the data collected in large PVC columns (14 kg wheat) could be used in stored grain bins to predict insect population dynamics.

The innate rate at the beginning of a population was the highest innate rate of all the tested conditions, even when the temperature was gradually increased. This was not expected because insects should have a high multiplication when temperature is increased. Before the innate rate became negative, the *r* values gradually decreased with an increasing storage time at any constant temperature in 13.8 to 14.5% MC wheat and for both *T. castaneum* and *C. ferrugineus*, regardless of the insect density. After the innate rate became negative (population decrease), the innate rate was able to increase; however, the rebounded innate rate could not increase more and gradually tended toward stability within an up-and-down range. This up-and-down rate was related to the carrying capacity under each tested temperature condition. The higher the grain temperature, the higher the carrying capacity and *r* value, but the *r* values at the carrying capacity were lower than the maximum *r* value. These results indicated that the population of stored grain insects was strongly influenced by the food sources and chemical environment, and a higher insect density could result in a lower and negative innate rate, except when the storage time for the wheat was less than 2 mos. A larger surface area and a higher proportion of cracked wheat could result in a higher innate rate of increase in *T. castaneum*. The open space might also be related to the chemical environment, even though insects can more easily move near the surface of the grain. This study also found the carrying capacity of both insect species decreased with decreasing temperature, and the innate rate for *T. castaneum* was more influenced by its density than that for *C. ferrugineus.*

## Figures and Tables

**Figure 1 insects-15-00441-f001:**
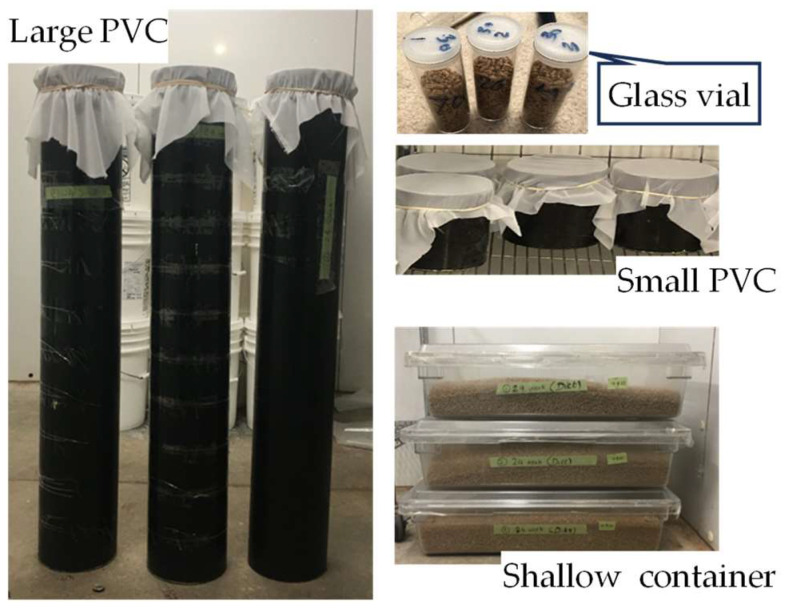
The grain containers or columns used to observe insect population dynamics under different storage conditions. In the graph, the glass vial had 50 mL inner volume, filled with 0.03 kg wheat; the small PVC column had 0.15 m diameter and was 15 cm high, total 2.6 L inner volume, filled with 2 kg wheat; large PVC column had 0.15 m diameter and was 102 cm high, total 18 L inner volume, filled with 14 kg wheat; and the shallow container was 46 cm long, 66 cm wide, and 15 cm high, filled with 14 kg wheat or wheat plus cracked wheat.

**Figure 2 insects-15-00441-f002:**
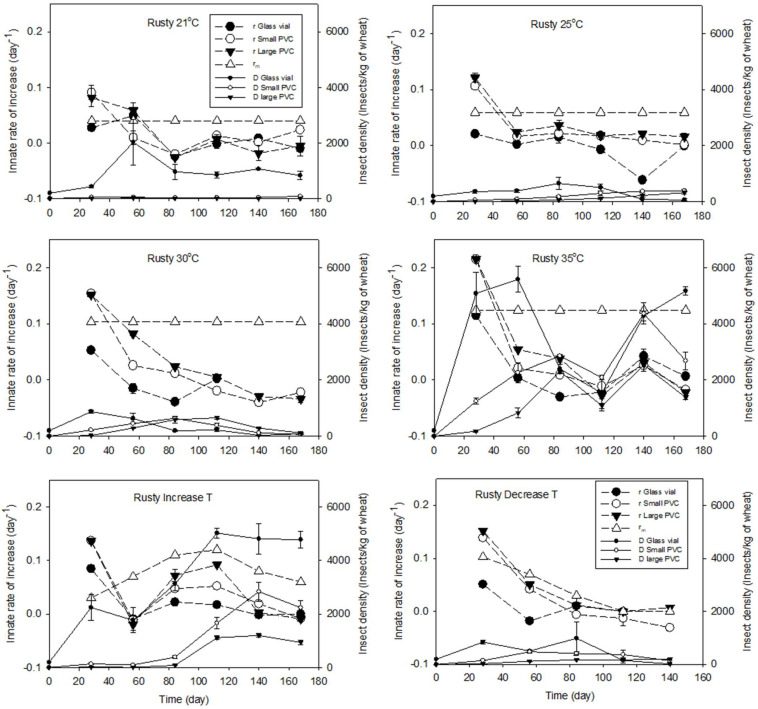
Innate rate of increase (*r_m_* and *r*) and insect density of *Cryptolestes ferrugineus* under different temperatures (21, 25, 30, 35, Increase T, and Decrease T) in wheat (13.8 to 14.5% moisture contents, about 60 to 70% relative humidity) at different storage times. In the legend, Rusty is the rusty grain beetle at the specified temperature. The *r* value is associated with the insects reared in Glass vial, Small PVC, and Large PVC. *r_m_* is the innate rate of increase reported by Smith [4]. D is the insect density. Decrease T = 30 °C in the first 4 wks, then decreased by 1 °C/wk, and Increase T = 21 °C in the first 2 wks, then increased by 1 °C/wk. The insect density was calculated using the number of adults plus offspring. Smith [4] reared insects in less than 0.5 kg of wheat flour plus 5% wheat germ (by mass) at 17.5 to 42.5 °C at 2.5 °C intervals.

**Figure 3 insects-15-00441-f003:**
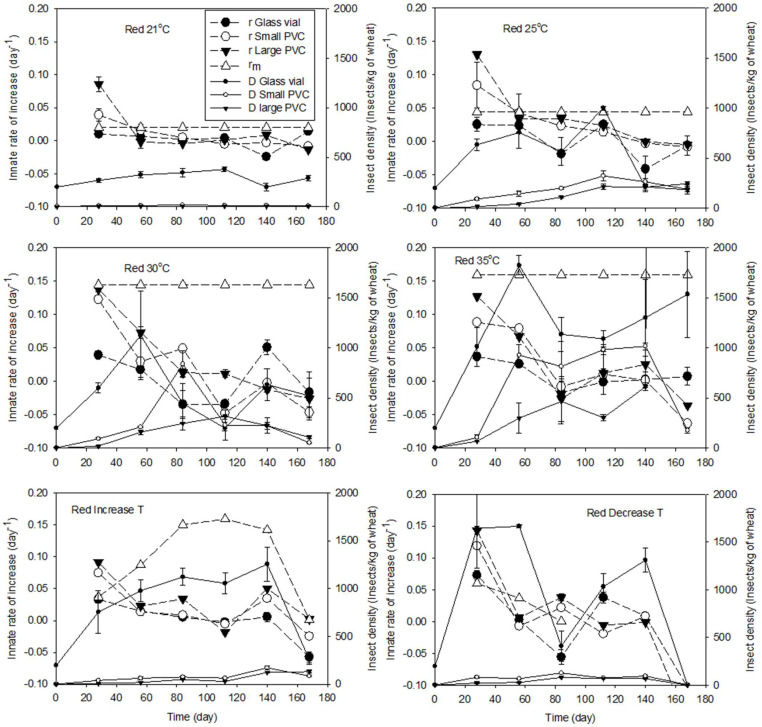
Innate rate of increase (*r_m_*_,_ and *r*) and insect density of *Tribolium castaneum* under different temperatures (21, 25, 30, 35, Increase T, and Decrease T) in wheat (13.8 to 14.5% moisture contents, about 60 to 70% relative humidity) at different storage times. In the legend, Red is the red flour beetle at the specified temperature, and the *r* value is associated with the insects reared in Glass vial, Small PVC, and Large PVC. *r_m_* is the innate rate of increase reported by White [7]. D is the insect density. Decrease T = 30 °C in the first 4 wks, then decreased by 1 °C/wk, and Increase T = 21 °C in the first 2 wks, then increased by 1 °C/wk. White [7] reared insects in less than 1 kg of wheat at 20 to 37.5 °C.

**Figure 4 insects-15-00441-f004:**
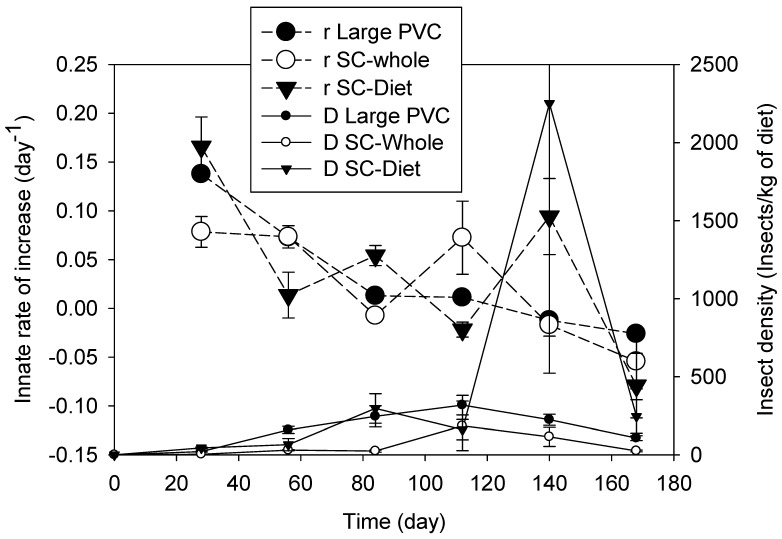
Innate rate of increase in (*r*) and insect density of *Tribolium castaneum* in different shapes of containers filled with whole wheat and diet at different storage times at 30 °C and 60 to 70% relative humidity. In the legend, *r* is the innate rate of increase; D is the insect density; Large PVC is the container with 0.15 m diameter and 102 cm high (18 L inner volume); SC is the shallow container 460 mm long, 660 mm wide, and 150 mm high; Whole is the whole wheat; and Diet is the diet (whole wheat and cracked wheat at 19:1 ratio by mass). The insect density was calculated using the number of adults plus offspring (larvae, pupae, and eggs).

**Table 1 insects-15-00441-t001:** Pearson’s correlation coefficient between the innate rate of increase and insect density of rusty grain beetle, *Cryptolestes ferrugineus*, and red flour beetle, *Tribolium castaneum,* under different temperatures (21, 25, 30, 35, Increase T, and Decrease T) in wheat (13.8 to 14.5% moisture contents, about 60 to 70% relative humidity) at different storage times.

Insects and Grain			21 °C	25 °C	30 °C	35 °C	Increase T ^a^	Decrease T ^a^
*C. ferrugineus*	Glass vial ^b^	A ^c^	−0.247	−0.380	−0.591 ^d^	−0.673	−0.390	−0.452
		A + O ^c^	−0.860	−0.463	−0.560	−0.797	−0.764	−0.993 *
	Small PVC ^b^	A ^c^	−0.187	−0.555	−0.655 ^d^	−0.650	−0.502	0.803
		A + O ^c^	−0.930 *	−0.690	−0.639	−0.816	−0.539	−0.917 *
	Large PVC ^b^	A ^c^	−0.610	−0.402	−0.768	−0.751	−0.546	−0.661
		A + O ^c^	−0.906 *	−0.535	0.800	−0.807	−0.597	−0.873
*T. castaneum*	Glass vial ^b^	A ^c^	−0.860 ^d^	−0.567	−0.574	−0.894 *	−0.722	−0.243
		A + O ^c^	−0.808	−0.615	−0.746	−0.901 *	−0.778	−0.891 *
	Small PVC ^b^	A ^c^	−0.662	−0.890 *	−0.614	−0.670	−0.757	−0.788
		A + O ^c^	−0.951 **	−0.953 **	−0.687	−0.930 **	−0.842 *	−0.807
	Large PVC ^b^	A ^c^	−0.503	−0.669	−0.723	−0.799	−0.525	−0.917 ** ^d^
		A + O ^c^	−0.968 **	−0.764	−0.893 *	−0.868 *	−0.620	−0.713

^a^ Decrease T = 30 °C in the first 4 wks, then decreased by 1 °C/wk, and Increase T = 21 °C in the first 2 wks, then increased by 1 °C/wk. ^b^ Insects were reared in small (glass vial with 50 mL inner volume filled with 0.03 kg wheat, referred to as Glass vial), medium (small PVC column with 0.15 m diameter and 15 cm high, total 2.6 L inner volume filled with 2 kg wheat, referred to as Small PVC), and large (large PVC column with 0.15 m diameter and 102 cm high, total 18 L inner volume filled with 14 kg wheat, referred to as Large PVC) grain volumes. ^c^ Correlation was conducted between the innate rate value and the adult density (A) or density of adults plus offspring (A + O). ^d^ Pearson’s correlation coefficients associated with the density of both adults and offspring were not higher than that associated with adult density only. * and ** Significant correlation at α < 0.05 and 0.001 levels, respectively.

**Table 2 insects-15-00441-t002:** Statistical test results among *r* values in different grain volumes at fluctuating temperatures and in one month of storage.

Statistic ^a^	Increase T ^b^		Decrease T ^b^	
	*C. ferrugineus*	*T. castaneum*	*C. ferrugineus*	*T. castaneum*
F ^b^	30.658	68.360	240.319	39.960
P ^b^	<0.001 **	<0.0001 ***	<0.0001 ***	<0.001 **

^a^ Comparison (Tukey’s test, df = 2) was conducted among the innate rate of increase values associated with different grain volumes. Insects were reared in small (glass vial with 50 mL inner volume filled with 0.03 kg wheat, referred to as Glass vial), medium (small PVC column with 0.15 m diameter and 15 cm high, total 2.6 L inner volume filled with 2 kg wheat, referred to as Small PVC), and large (large PVC columns with 0.15 m diameter and 102 cm high, total 18 L inner volume filled with 14 kg wheat, referred to as Large PVC) grain volumes. ^b^ Decrease T = 30 °C in the first 4 wks, then decreased by 1 °C/wk, and Increase T = 21 °C in the first 2 wks, then increased by 1 °C/wk. The column shows the F and *p* values of Tukey’s test. ** and *** Significant difference in the test at α < 0.001 or 0.0001 level, respectively.

**Table 3 insects-15-00441-t003:** Carrying capacity of *C. ferrugineus* and *T. castaneum* at different temperatures, in different wheat columns, and at 60 to 70% relative humidity.

Insects and Grain			21 °C ^a^	25 °C ^a^	30 °C ^a^	35 °C ^a^
*C. ferrugineus*	Glass vial ^b^	A ^c^	588.9 ± 11.1 **B**	522.2 ± 107.2 **B**	188.9 ± 35.3 **B**	255.6 ± 66.4 **B**
		A + O ^cd^	1006.7 ± 33.3 **e**	655.6 ± 223.1 **ef**	222.2 ± 48.4 **f**	4277.8 ± 173.6 **g**
	Small PVC ^b^	A ^c^	17.3 ± 2.6 **C**	57.3 ±8.8 **C**	381.0 ± 43.4 **C**	1070.8 ±477.0 **B**
		A + O ^cd^	41.5 ± 5.8 **e**	182.2 ± 38.7 **e**	638.3 ± 51.8 **e**	4361.5 ± 383.2 **f**
	Large PVC ^b^	A ^c^	2.1 ± 0.1 **C**	14.1 ± 1.2 **C**	443.28 ± 50.3 **C**	884. 4± 76.4 **B**
		A + O ^cd^	14.8 ± 6.0 **e**	69.7 ± 7.8 **e**	659.7 ± 55.1 **f**	2535.7 ± 224.9 **g**
*T. castaneum*	Glass vial ^b^	A ^c^	322.2 ± 29.4 **B**	244.4 ± 22.2 **B**	700.0 ± 157.5 **B**	1066.7 ± 171.1 **B**
		A + O ^cd^	377.8 ± 22.2 **e**	755.6 ± 156.7 **ef**	1122.2 ± 441.5 **ef**	1822.2 ± 96.9 **f**
	Small PVC ^b^	A ^c^	5.2 ± 0.2 **C**	217.5 ± 38.5 **BC**	562.5 ± 129.4 **B**	256.5 ± 28.4 **C**
		A + O ^cd^	16.2 ± 1.2 **e**	323.2 ± 46.6 **e**	836.8 ± 129.4 **fg**	925.2 ± 107.7 **g**
	Large PVC ^b^	A ^c^	0.4 ± 0.0 **C**	118.2 ± 8.4 **C**	245.7 ±35.5 **B**	247.9 ± 32.0 **C**
		A + O ^cd^	5.2 ± 0.2 **e**	110.0 ±8.4 **f**	317.9 ± 62.7 **f**	612.7 ± 35.9 **f**

^a^ Different capital letters in the column after the carrying capacity values indicate a significant difference at the same temperature (in the column) and in different wheat columns (Tukey’s test, N = 3) at α < 0.05 level. The comparison among the total insect numbers had the same result as that among the adult number. ^b^ Insects were reared in small (glass vial with 50 mL inner volume filled with 0.03 kg wheat, referred to Glass vial), medium (small PVC column with 0.15 m diameter and 15 cm high, total 2.6 L inner volume filled with 2 kg wheat, referred to as Small PVC), and large (large PVC column with 0.15 m diameter and 102 cm high, total 18 L inner volume filled with 14 kg wheat, referred to as Large PVC) grain volumes. ^c^ A = adult density, A + O = density of adults plus offspring. ^d^ Different lowercase letters in the row after the carrying capacity indicate a significant difference in the same wheat column (in the row) at different temperatures (Tukey’s test, N = 3) at α < 0.05 level. The comparison among the total insect numbers had the same result as that among the adult number.

## Data Availability

No new data were collected in this study, and the calculated data were reported in the article.

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
