# Peer review of "The Innate Rates of Increase in Two Common Stored Grain Insects under Different Grain Storage Conditions and Times"

_insects, 2024, doi:10.3390/insects15060441_

Round 1
Reviewer 1 Report
Comments and Suggestions for Authors
MS ID: insects-3010815
General comments:
The article titled “Innate rate of increase of two common stored grain insects under different grain storage conditions and times” by the author Fuji Jian, is a specific scientific study which contributes to the knowledge in this field. I suggest the editor to accept this article for publishing in the Insects journal, after following corrections.
Specific notes:
Title is adequate
Simple Summary, Abstract and Keywords are OK.
Introduction
The text between Lines 37-48 should be confirmed with appropriate literature sources.
Materials and Methods
In this section should be explained all mathematical equations, the meaning of parameters and their units, better and with more details than in Introduction.
According to this section in this work no experiments are done, only used values from previous experiments and put in new calculations. Therefore, add at least one demonstrative example for each equation how are parameters determined and the calculation performed.
Results
There are no graphic representation of the dependance of parameter „α“ to parameters „r“ and „N“, and the dependance of parameter „R“ to parameter „r“ and the dependance of parameter „α“ to „k“.
Explain the meaning of „k“ (carry capacity) and its units, because according to the text in Discussion it is very important for Conclusion and purpose of the work. Therefore, this equation should be mentioned and explained in Introduction and Materials and methods and not in this section for the first time.
Tables and Figures are adequate.
Discussion
The sentence in Line 475 is redundant, because parameter „α“ is obviously dependent of „r“ and „k“, according to equation and this should be explained earlier.
Explain how is parameter „α“: „equivalent of the intraspecific competition...“.
Explain how can unit for „α“ be „kg of grain/day or insect“?
The explanation for temperature influence is not clear and not according to mathematical equations. According to exponential equations parameters „r“ and „t“ are inversely proportional, so, it is not strange that with „t“ increase, „r“ is decreasing.
The total insect density should be mentioned in previous sections of the MS in the form of better explanation and some graphic curve that shows the influence of insect density to population growth.
The influence of „grain bulk“ to „r“ should also be explained earlier and represented in some figure.
Conclusion
This section should be rewritten avoiding repetition of the results and at the end of the text add short prediction of future investigations and interests.
Reference
In some references write full names of the authors.
Author Response
Title is adequate
Simple Summary, Abstract and Keywords are OK.
Introduction
The text between Lines 37-48 should be confirmed with appropriate literature sources.
Response. A reference has been added.
Materials and Methods
In this section should be explained all mathematical equations, the meaning of parameters and their units, better and with more details than in Introduction.
According to this section in this work no experiments are done, only used values from previous experiments and put in new calculations. Therefore, add at least one demonstrative example for each equation how are parameters determined and the calculation performed.
Response. An example was added. The meaning of the r and rm has been added.
Results
There are no graphic representation of the dependance of parameter „α“ to parameters „r“ and „N“, and the dependance of parameter „R“ to parameter „r“ and the dependance of parameter „α“ to „k“.
Explain the meaning of „k“ (carry capacity) and its units, because according to the text in Discussion it is very important for Conclusion and purpose of the work. Therefore, this equation should be mentioned and explained in Introduction and Materials and methods and not in this section for the first time.
Response. The carrying capacity was introduced in the introduction and method. The symbol “k” was added at the first introduction. The equation 5 was developed from Eq. 4. Therefore, all the meaning of the symbols in Eq. 5 had been explained.
Tables and Figures are adequate.
Discussion
The sentence in Line 475 is redundant, because parameter „α“ is obviously dependent of „r“ and „k“, according to equation and this should be explained earlier.
Response. I disagree with this comment because the relationship should be clearly stated, and the following comments suggested by the same reviewer are contradicted to this comment.
Explain how is parameter „α“: „equivalent of the intraspecific competition...“. Explain how can unit for „α“ be „kg of grain/day or insect“?
Response. The study published by Mallet 2012 provided the detailed information about this equation and the meaning of each parameter. Considering my study did not focus on this equation development and the complexity of this topic, I added the reference in the revised version, so the interested reader can read it.
The explanation for temperature influence is not clear and not according to mathematical equations. According to exponential equations parameters „r“ and „t“ are inversely proportional, so, it is not strange that with „t“ increase, „r“ is decreasing.
Response. The reviewer made a mistake because here the “t” is time, not temperature.
The total insect density should be mentioned in previous sections of the MS in the form of better explanation and some graphic curve that shows the influence of insect density to population growth.
Response. This was explained in the section 2.2. Calculation of r and carrying density.
The influence of „grain bulk“ to „r“ should also be explained earlier and represented in some figure.
Response. This was presented in figured because glass vial, small PVC, and large PVC used for testing insect population dynamics have different sizes of grain bulk.
Conclusion
This section should be rewritten avoiding repetition of the results, and at the end of the text add short prediction of future investigations and interests.
Response. I agree with this comment. However, absolutely no overlap is impossible because conclusion should confirm the results. Adding a short prediction is not a good idea because it is not a conclusion of this study.
Reference
In some references write full names of the authors.
Response. It is corrected.
Reviewer 2 Report
Comments and Suggestions for Authors
The author presents a new look at previous work on innate rate of increase and carrying capacity using previously published work. A careful look at what was previously published vs what is a new calculation is needed, but is mostly clear. This distinction is most clear in the discussion. The work adds to the growing literature on population growth and ecology in grain bins.
Specific comments
Line 45: Due to the above reason.
Line 57: Insect populations
Line 59: but provides
Line 61: What does (decimal) mean? Could you instead provide a range (0 to 1)? Please clarify
Lines 60, 67, 72, 80: Make reference to your equations before you present them. For example, you could write "...by calculating its instantaneous rate of increase per unit time (Eq. 1)..." in lines 57-58.
Line 96: Maybe somewhere in this paragraph or the previous section, you should explicitly state what a value of r means. For example, if r > 0 then it means what?
Figure 1: Are these images from previously published literature? If so, you should cite them.
Lines 234-235: I'm not sure what the 0C detail means? The grain was cooled after 5 months? Did that stop the population so they could make their counts? More details here.
Section 2.3: What program did you use to do your statistics?
Line 243: What normality test did you use?
Line 276: Phrasing here is odd. Please revise.
Figure 2: On the graph, consider replacing "Rusty" with the scientific name as referred in the legend.
Line 356: Person = Pearson
Line 252: Do you need to number this line 4) ?
Comments on the Quality of English LanguageMinor English check is necessary. Some comments above.
Author Response
In some references write full names of the authors.
Response. It is corrected.
Reviewer 2
The author presents a new look at previous work on innate rate of increase and carrying capacity using previously published work. A careful look at what was previously published vs what is a new calculation is needed, but is mostly clear. This distinction is most clear in the discussion. The work adds to the growing literature on population growth and ecology in grain bins.
Specific comments
Line 45: Due to the above reason. Line 57: Insect populations. Line 59: but provides.
Response. Thanks for the suggestion. All were corrected.
Line 61: What does (decimal) mean? Could you instead provide a range (0 to 1)? Please clarify.
Response. Decimal is decimal number. This is commonly used for unit. Therefore, no correction, but its meaning was added when the r and rm were introduced.
Lines 60, 67, 72, 80: Make reference to your equations before you present them. For example, you could write "...by calculating its instantaneous rate of increase per unit time (Eq. 1)..." in lines 57-58.
Response. It has been changed.
Line 96: Maybe somewhere in this paragraph or the previous section, you should explicitly state what a value of r means. For example, if r > 0 then it means what?
Response. It has been added.
Figure 1: Are these images from previously published literature? If so, you should cite them.
Response. The previous published literatures have not this image.
Lines 234-235: I'm not sure what the 0C detail means? The grain was cooled after 5 months? Did that stop the population so they could make their counts? More details here.
Response. The reviewer’s assumption is correct. This information has been added.
Section 2.3: What program did you use to do your statistics?
Response. The information related to the used software was added.
Line 243: What normality test did you use?
Response. Shapiro–Wilk test, Kolmogorov–Smirnov test, and mean with SD. This information has been added.
Line 276: Phrasing here is odd. Please revise.
Response. It has been rephrased.
Figure 2: On the graph, consider replacing "Rusty" with the scientific name as referred in the legend.
Response. This is a good suggestion, but I did not change it because “Rust” and “Red” is easily referred (because it is short. In some figure, space is an issue) and these two symbols were explained in the figure caption.
Line 356: Person = Pearson
Response. It has been corrected.
Line 252: Do you need to number this line 4) ?
Response. It has been changed.
Reviewer 3 Report
Comments and Suggestions for Authors
Reviewer Comment and Recommendation for Manuscript ID: Insects-3010815-peer-review-v1
Title: Innate rates of increase of two common stored grain insects under different grain storage conditions and times
Specific comments and suggestions:
Simple Summary
L.10: Change “comparation” to “comparison”
Abstract
L.20: Replace “r (innate rate of increase)’ with “innate rate of increase (r)”
L.22: “....by using a new suggested method...” What method? Add it
L.24: Replace the “small, large, and shallow” with the volume or dimensions
Keywords
L.33-34: Author provided “Keyphrases” instead of “keywords”. Please, provide about 5 keywords that are not part of the “Title”
Introduction
L.38: “... because they do not need energy to regulate their body temperature”; this is ambiguous. Both Ectotherm and Endotherm required energy for thermoregulations, however, the source (internally generated or relying on the environment). Therefore, reword lines 37-38.
L.49: Spell-out “wk”
L.105: Change “might” to “can”. Repeat this throughout the text (L.115 and others)
Materials and Methods
L.152: Add Order and Family names after the author’s name for both insect species.
L.163: Change “ther” to “the”
L.163: How many individuals were in each “patch”? Were patches inoculated with single or mixed species of the insects?
L.156: “T-decrease” what is it?
L.154-158: “The data reported....wet basis” is not clear. Revise it
L.159: Change “less than 0.5 kg” to “0.03 kg”. Refer to L.190
L.169: Change “the same as” to “similar to”
L.175: The bracket content is not clear; revise
L.189: Pictures are of low quality and labels impeded the view of the grain containers. Provide a better figure.
L.205: Red flour beetles are secondary pests (external feeders) that will not do well on whole wheat (L.206); could that have impacted negatively on the quality of data (larvae mortality)?
Results
L.254. “(about 1 mon)”; provide the exact duration in weeks used in the experiment
L.257: Change “Fig,2” to “Fig.2”
L.265: Delete “the” in “was the similar”
L.316: Fig. 3 “25 to 65% RH”; show T. castaneum survive below 43% RH for about 30+ wk? Most stored product insects rarely survive long term (> 40 d) dehumidification (<43% RH) at the temperature range (20 to 37.5).
L.335: Fig. 4. Add the RH after the temperature value
L.339: “plus offspring” What developmental stage? Larvae, pupae or both?
L.363: Change “Increse” to “Increase)
L.365: “T-increase” T increase” Increase T, etc.; be consistent throughout the text.
Also, “Decrease T=30 ÌŠC in the first 4 wk” is confusing. What was the temperature at week zero (0 wk: initial temperature)? With few lines in the text, clearly explain this in the “Materials and Methods” section by providing the initial and final temperature values for each increasing or decreasing temperature and at what wk?
L.398: “mon”? Spell out or use “mo” Make this correction throughout the text
L.417: “moth”?
L.423-436 and others: Maintain similar font type throughout the text
Discussion
L.440: Provide reference (s) after “Malthusian parameter”
L,443-444: “Innate .....density)” not clear; modify it
L.448: Spell out “mon”; either use “month” or “mo” not “mon”= Monday
L.466-471: Define all terms in the model (s)
L.495-496: Not clear; revise
L.513-514: “This difference....temperatures” not clear; revise
Conclusions
L.540, 541 & 561: Correct “Castaneum”
L.559: “mon”?
Comments on the Quality of English LanguageModerate editing is required.
Author Response
Simple Summary
L.10: Change “comparation” to “comparison”
Abstract
L.20: Replace “r (innate rate of increase)’ with “innate rate of increase (r)”
Response. It has been changed.
L.22: “....by using a new suggested method...” What method? Add it
Response. The method has been added.
L.24: Replace the “small, large, and shallow” with the volume or dimensions
Response. The mass of the wheat has been added.
Keywords
L.33-34: Author provided “Key phrases” instead of “keywords”. Please, provide about 5 keywords that are not part of the “Title”
Response. It has been changed.
Introduction
L.38: “... because they do not need energy to regulate their body temperature”; this is ambiguous. Both Ectotherm and Endotherm required energy for thermoregulations, however, the source (internally generated or relying on the environment). Therefore, reword lines 37-38.
Response. It has been rephrased as “they need much less energy to regulate their body temperature”.
L.49: Spell-out “wk”
Response. It has been changed.
L.105: Change “might” to “can”. Repeat this throughout the text (L.115 and others)
Response. It has been changed.
Materials and Methods
L.152: Add Order and Family names after the author’s name for both insect species.
Response. It has been added.
L.163: Change “ther” to “the”
Response. It has been changed.
L.163: How many individuals were in each “patch”? Were patches inoculated with single or mixed species of the insects?
Response. The following information has been added: Single insect species was introduced in each patch and insect number changed during the test period (maximum 217 d).
L.156: “T-decrease” what is it?
Response. It has been explained in the bracket.
L.154-158: “The data reported....wet basis” is not clear. Revise it
Response. It has been rephrased and more information has been added.
L.159: Change “less than 0.5 kg” to “0.03 kg”. Refer to L.190
Response. The “less than 0.5 kg” was correct because the original publication stated in this way.
L.169: Change “the same as” to “similar to”
Response. It has been changed.
L.175: The bracket content is not clear; revise
Response. The bracket had been removed and new statement had been added.
L.189: Pictures are of low quality and labels impeded the view of the grain containers. Provide a better figure.
Response. Pictures were retaken. Labels were rearranged.
L.205: Red flour beetles are secondary pests (external feeders) that will not do well on whole wheat (L.206); could that have impacted negatively on the quality of data (larvae mortality)?
Response. This has been discussed in the section before Fig. 2 in the current MS, and the original publication already discussed this topic.
Results
L.254. “(about 1 mon)”; provide the exact duration in weeks used in the experiment
Response. It has been changed.
L.257: Change “Fig,2” to “Fig.2”
Response. It has been changed.
L.265: Delete “the” in “was the similar”
Response. It has been deleted.
L.316: Fig. 3 “25 to 65% RH”; show T. castaneum survive below 43% RH for about 30+ wk? Most stored product insects rarely survive long term (> 40 d) dehumidification (<43% RH) at the temperature range (20 to 37.5).
Response. The 25 is a mistake. Thanks for pointing out. It was deleted.
L.335: Fig. 4. Add the RH after the temperature value.
Response. I has been added.
L.339: “plus offspring” What developmental stage? Larvae, pupae or both?
Response. Includes larvae, pupae and eggs. The information has been added.
L.363: Change “Increse” to “Increase)
Response. It has been changed.
L.365: “T-increase” T increase” Increase T, etc.; be consistent throughout the text.
Response. It has been changed globally.
Also, “Decrease T=30 ÌŠC in the first 4 wk” is confusing. What was the temperature at week zero (0 wk: initial temperature)? With few lines in the text, clearly explain this in the “Materials and Methods” section by providing the initial and final temperature values for each increasing or decreasing temperature and at what wk?
Response. It has been changed.
L.398: “mon”? Spell out or use “mo” Make this correction throughout the text
Response. It has been changed.
L.417: “moth”?
Response. It is month. It has been changed.
L.423-436 and others: Maintain similar font type throughout the text.
Response. It has been changed.
Discussion
L.440: Provide reference (s) after “Malthusian parameter”
Response. The reference has been added.
L,443-444: “Innate .....density)” not clear; modify it
Response. It has been rephrased.
L.448: Spell out “mon”; either use “month” or “mo” not “mon”= Monday
Response. It has been changed.
L.466-471: Define all terms in the model (s)
Response. All the terms have been defined.
L.495-496: Not clear; revise
Response. It has been rephrased.
L.513-514: “This difference....temperatures” not clear; revise
Response. It has been rephrased.
Conclusions
L.540, 541 & 561: Correct “Castaneum”.
Response. It has been corrected.
L.559: “mon”?
Response. It has been corrected.